# Development of Novel Fluorinated Polyphenols as Selective Inhibitors of DYRK1A/B Kinase for Treatment of Neuroinflammatory Diseases including Parkinson’s Disease

**DOI:** 10.3390/ph16030443

**Published:** 2023-03-15

**Authors:** Gian Luca Araldi, Yu-Wen Hwang

**Affiliations:** 1Avanti Biosciences, Inc., 3210 Merryfield Row, San Diego, CA 92121, USA; 2New York State Institute for Basic Research in Developmental Disabilities, Department of Molecular Biology, 1050 Forest Hill Road, Staten Island, NY 10314, USA

**Keywords:** DYRK1A, Parkinson’s disease, polyphenol, EGCG, inflammation, LPS, MPTP

## Abstract

Natural polyphenol derivatives such as those found in green tea have been known for a long time for their useful therapeutic activity. Starting from EGCG, we have discovered a new fluorinated polyphenol derivative (**1c**) characterized by improved inhibitory activity against DYRK1A/B enzymes and by considerably improved bioavailability and selectivity. DYRK1A is an enzyme that has been implicated as an important drug target in various therapeutic areas, including neurological disorders (Down syndrome and Alzheimer’s disease), oncology, and type 2 diabetes (pancreatic β-cell expansion). Systematic structure–activity relationship (SAR) on trans-GCG led to the discovery that the introduction of a fluoro atom in the D ring and methylation of the hydroxy group from para to the fluoro atom provide a molecule (**1c**) with more desirable drug-like properties. Owing to its good ADMET properties, compound 1c showed excellent activity in two in vivo models, namely the lipopolysaccharide (LPS)-induced inflammation model and the 1-methyl-4-phenyl-1,2,3,6-tetrahydropyridine (MPTP) animal model for Parkinson’s disease.

## 1. Introduction

We seek to develop a potent and selective negative allosteric modulator of “Dual-specificity tyrosine-(Y)-phosphorylation Regulated Kinase-1A” (DYRK1A) to treat neuroinflammatory and neurodegenerative diseases such as Parkinson’s disease (PD). PD is the second most common neurodegenerative disorder, affecting 1% of the global population above the age of 60 [1]. PD is identified as the fastest-growing neurological disease in terms of prevalence, disability, and deaths [2], and the number of PD patients is expected to double in the next 30 years. Hallmarks of PD include the appearance of Lewy bodies (LB), aggregates of misfolded protein, and dopaminergic neuronal loss in the substantia nigra pars compacta, which lead to characteristic motor symptoms [3].

DYRK1A is a ubiquitous enzyme that is overexpressed in Down syndrome patients through gene-dosage effects of trisomy chromosome 21 [4]. DYRK1A inhibition offers an attractive approach for the treatment of PD pathologies because of its specific enzymatic activity and broad involvement in anti-inflammatory pathways. Several pieces of evidence point to a role for DYRK1A in PD pathogenesis. Genome-wide association studies have revealed that DYRK1A is a risk factor for PD [5]. The DYRK1A rs8126696 T allele was associated with early onset in a cohort of 297 Chinese patients with PD [6]. An additional study in the Chinese Han population identified the TT genotype derived from SNP rs8126696 of the DYRK1A gene as a possible risk factor for developing sporadic PD, especially for men [7]. DYRK1A may influence the behavior of parkin, the protein product of the first gene known to cause autosomal recessive familial PD [8]. DYRK1A directly phosphorylates parkin at Ser131 in vitro, which inhibits parkin’s E3 ubiquitin ligase activity and, consequently, its neuroprotective function in dopaminergic SHSY5Y cells exposed to 6-hydroxydopamine [9]. In PD, α-synuclein aggregates often contain and sequester septin 4 (SEPTIN4), a polymerizing scaffold protein. Septin 4 was identified as a DYRK1A-binding partner in a yeast two-hybrid screen, which colocalizes with DYRK1A in mouse neurons. DYRK1A phosphorylation of septin 4 is inhibited by harmine [10], suggesting a role for DYRK1A in the health of dopaminergic neurons [11]. Further evidence suggests that DYRK1A expression is increased in PD and Pick disease [12].

DYRK1A inhibitors can also reduce inflammation by targeting pathways including GFAP, STAT, T_reg_/Th17 differentiation [13], and NF-κB [14]. DYRK1A apparently controls the branch point of CD4 T_reg_ and Th17 cell differentiation. Inhibition of DYRK1A could tilt the balance to T_reg_, leading to a reduction in inflammation. Likewise, inhibition of DYRK1A could diminish neuroinflammation elicited by LPS, probably through suppression of the TLR4/NF-κB pathway [15]. It has been shown that DYRK1A phosphorylates cyclin D1, leading to a decrease in p21 in the cells and, ultimately, to reduced expression of Nrf2, a transcription factor that induces the expression of genes involved in antioxidant pathways, which reduce ROS levels. DYRK1A inhibitors can potentiate the neuroprotective p21-Nrf2 pathway and contribute to neuronal survival by lessening proinflammatory cytokine production caused by neuroinflammation [16]. Taken together, because of DYRK1A’s interactions with several factors involved in PD and its multiple positions in regulating inflammation, we intend to test the therapeutic hypothesis that by reducing DYRK1A activity, it is possible to intervene in PD pathogenesis and to slow or stop disease progression.

DYRK kinases, which are activated by autophosphorylation of tyrosine residues present in a sequence of the activation loop, phosphorylate serine/threonine residues in the substrate [17]. In mammals, five subtypes, i.e., DYRK1A, 1B, 2, 3, and 4, have been identified. These subtypes are divided into two groups: class I, which includes DYRK1A and DYRK1B, and class II, comprising DYRK2, DYRK3, and DYRK4. DYRKs in the same class have high sequence homology and, therefore, may show similar affinity toward inhibitors [18].

To date, most emphases have been placed on developing ATP-competitive DYRK1A inhibitors such as harmine, a β-carboline alkaloid, and many others (Figure 1) [19]. Most of these compounds inhibit both 1A and 1B with the same strength, and few selective derivatives have been discovered to date, such as AZ-191, a selective DYRK1B inhibitor [20]. An alternate attractive DYRK1A inhibitor is the natural product epigallocatechin gallate (EGCG), which combines activity against DYRK1A with antioxidant activity thanks to its polyphenolic groups. The advantage of this class of DYRK1A inhibitors vs. current therapies is the multiple mechanisms of action of such inhibitors, with a potentially superior safety profile. EGCG has been shown to function as a DYRK1A allosteric inhibitor with an in vitro IC_50_ of around 300–400 nM [21,22]. Previous studies have reported that EGCG prevented MPTP-induced loss of dopaminergic neurons in the substantia nigra, which was concomitant with a depletion in striatal dopamine and tyrosine hydroxylase (TH) protein levels. Another study demonstrated that the protective effects of EGCG in the MPTP mouse model of PD were realized via inhibition of neuronal nitric oxide synthase in the substantia nigra [23]. Moreover, EGCG has immunomodulatory effects in many disease models, including nervous system disease. Zhou et al. (2018) recently demonstrated that EGCG has neuroprotective effects in an MPTP-induced PD mice model, which may be exerted by modulating the peripheral immune response [24].

Earlier in the project, we observed that trans-catechin derivatives are about twofold more potent than the corresponding cis derivatives in inhibiting DYRK1A [25]. To our surprise, introducing one or two fluoros in the ortho position of the B or D rings led to a dramatic increase in anti-DYRK1A activity. For example, **1b** (Table 1, GCG-2”,6”diF) showed about an eightfold improvement in activity compared to EGCG and is one of the most potent DYRK1A inhibitors discovered to date [26]. Despite its promising activity, the development of EGCG has been hampered by its poor pharmacokinetic profile. Similarly, pharmacokinetic (PK) studies performed in mice using GCG monofluoro or difluoro derivatives showed poor bioavailability profiles similar to that of EGCG (compound **1b**, Table 2). It is well known that catechin methylation leads to a molecule such as natural EGCG-3”OMe, which is highly abundant in Benifuuki tea and characterized by much-improved bioavailability [27], although usually with reduced biological activity. Here, we describe the effects of alkylating the hydroxyl group in the meta position of the D ring of our fluorinated GCG derivatives. These molecules were designed with the aim of combining high activity against DYRK1A and good bioavailability.

## 2. Results

### 2.1. Synthesis of Compounds

As previously described [26], we developed a robust four-step process for synthesizing D-ring derivatives from readily available natural epigallocatechin (EGC) (Figure 1, as adapted from [26]).

This procedure was used to prepare a series of D-ring derivatives with modifications on the metahydroxy group, together with ortho and/or fluoro atom(s) (compounds **1a–h** in Table 1) for the current study. Briefly, EGC was first isomerized to yield gallocatechin (GC), which was then converted to alcohol intermediate 2 by selective benzylation of the phenolic groups. Intermediate 2 was esterified using benzylic acid derivatives **3a–h** to produce ester **4a–h**, followed by catalytic hydrogenation to provide the desired derivatives (**1a–h**). The detailed procedures for each step and for the preparation of various acids (**3a–h**) are described in the Section 4.

### 2.2. Overview of Activities and SAR

We previously observed that having only one OH or capping the para OH in the D ring reduces activity. However, the introduction of fluorine in ortho resulted in a marked increase in activity. Because methylation of metaphenolic alcohol generates derivatives characterized by considerably improved pharmacokinetic properties, we studied the introduction of substituents on the metahydroxy group, together with fluoro atom(s) in the ortho position. The activity of the new compounds, together with their selectivity against other DYRK subtypes, is reported in Table 1. As expected, methylation of the metaphenol (GCG-3”OMe) reduced the activity of parent GCG by half. However, fluorination in position 2” yielded compound **1c**, which is actually more potent than GCG (73 nM vs. 121 nM). The introduction of fluorine in ortho to the methoxy group (compound 1d) or two fluorine atoms (compound **1e**) did not improve activity. Replacement of the methyl group with a difluoro methyl (compounds **1f** and **1g**) or isopropyl (compound **1h**) is deleterious for the overall activity. The best compounds were tested for their selectivity against the closely related DYRK1B and DYRK2. As described in Table 1, these molecules showed little or no selectivity vs. DYRK1B and were about four to five times more selective than DYRK2.

Systemic exploration of the various combinations led to the discovery of compound **1c** (GCG-2”F,5”OMe), which has methoxy and fluorine groups para to each other. This compound was found to be the most potent methoxy derivative in our functional assay, with an IC_50_ of 73 nM. The inhibitory potency of **1c** was subsequently examined with varying concentrations of ATP as previously described [28]. Like its parent compound, EGCG, the inhibitory activity was minimally affected by ATP up to 0.8 mM. Thus, we conclude that **1c** is a non-ATP-competitive inhibitor.

### 2.3. Pharmacokinetic Studies

To overcome the generally poor PK profile of catechins, we developed a proprietary formulation that allows for the delivery of these drugs in high concentrations for oral (PO) or intranasal (IN) use [25]. Our liquid formulation is based on the solubilizer and penetration enhancer (2-hydroxypropyl)-β-cyclodextrin (HP-β-CD), PEG-400, and water. This vehicle allows us to achieve high drug concentration in dosing solution (up to 20% *w*/*w*) when needed. PK studies were conducted for **1c** in adult male and female C57BL/6 mice (*n* = 3/sex/time point) using (1) a single IV dose (5 mg/kg), (2) an IN dose (60 mg/kg, 5 µL of a clear solution/nostril/twice), and (3) a single PO dose (100 mg/kg). We analyzed the concentration of the drug in several tissues, including the brain, lungs, liver, and plasma. We observed that this class of molecules is stable in circulation; however, when the tissues are harvested for bioanalytical work, the drug undergoes rapid degradation, probably because of the direct contact with oxygen and metals, which catalyzes its rapid degradation/oxidation. We were able to stabilize the drug in blood by the immediate addition of ascorbic acid and TCEP [29]; however, the same was not possible with solid organs such as the brain and liver, which need to be collected through surgery and homogenized before any stabilizing agent can be added. Therefore, tissue PK data tend to vary.

Taking into consideration the above observations, when **1c** was delivered via PO, its absolute bioavailability in plasma was about 16%, while via IN, it was completely bioavailable (100%). Direct comparison with EGCG using oral dosing (100 mg/kg) shows that **1c** is characterized by a much-improved oral bioavailability (F_1c_ = 16 vs. F_EGCG_ = 2, Table 2). Using the PO route, we also achieved a high and stable exposure in all the analyzed tissues, including the brain (Table 2), which allows us to achieve a theoretical cellular exposure of about 3 times the IC_50_ of inhibiting DYRK1A. Interestingly, despite a higher plasma exposure, IN seems to deliver a lower amount of drug into tissues such as the brain and liver. Despite the high variability, as previously discussed, this lower exposure when using the IN route compared to PO dosing seems to be consistent and was also confirmed by the drug efficacy in animal models, as shown below. Since when dosing PO, we observed a higher concentration in conjugated drugs (preliminary observation not shown), we hypothesized that this is the form that might be responsible for the increased bioavailability in tissues. Evidence from recent publications shows that polyphenols and flavonoids can generally be conjugated at the intestinal level and delivered to the tissues of interest, where they are freed-up from the conjugate moiety and able to exert their pharmacological action. Tu et al. showed that the disposition of many oral phenolics is mediated by intestinal glucuronidation and hepatic recycling in a new disposition mechanism called ‘Hepatoenteric Recycling (HER)”, where the intestine is the metabolic organ and the liver is the recycling organ [30]. In their report, Perez-Vizcaino et al. indicated that “glucuronidated derivatives transport quercetin and its methylated form, and deliver to the tissues the free aglycone, which is the final effector” [31]. A similar scenario can be envisioned here in which compound **1c**, when administered via the PO route, is conjugated at the intestinal level and transported as glucuronide to the brain through the action of specific transport proteins of the blood–brain barrier (BBB) such as OATP1B; then, in the brain, the glucuronide is removed thanks to the action of selected glucuronidase enzymes. The same would not be possible when the drug is delivered IN, since it would bypass the intestinal tract, which is the main conjugation site. Studies are currently being undertaken to validate this hypothesis.

Finally, we tested the PK profile of our best in vitro inhibitor, **1b**. Unfortunately, in this case, the bioavailability was much lower than that of **1c**. These results confirmed what was already reported in the literature, in which catechin without the methyl group in the D ring showed a much lower bioavailability. Overall, based on the in vitro and in vivo data, we chose compound **1c** for our proof-of-concept efficacy animal studies reported below. To further confirm the PK results, we selected both PO and IN routes for the efficacy studies.

### 2.4. Pharmacological Studies

Neuroinflammation is a common feature shared by several neurodegenerative disorders and is implicated in the advancement of neurodegeneration. Dysregulated microglial activation causes neuroinflammation and has long been considered a treatment target in therapeutic strategies [32]. As outlined in the Introduction, DYRK1A regulates multiple inflammatory signaling pathways. DYRK1A inhibitors have been shown to suppress neurodegeneration caused by central and peripheral inflammation; thus, they may be broadly applicable for the treatment of various inflammatory conditions. Based on the above findings, we studied the efficacy of **1c** in the LPS-induced inflammation model and the MPTP PD model.

#### 2.4.1. LPS-Induced Inflammation Studies

Neuroinflammation is an important factor contributing to cognitive impairment and neurodegenerative diseases such as PD, and the administration of LPS is frequently used to study neuroinflammation-associated diseases in mice [33]. A recent study showed that DYRK1A inhibition reduced neuroinflammation, decreased microglial activation, and attenuated inflammation-induced neuronal damage in an LPS-induced neuroinflammatory model. Inhibition of DYRK1A attenuated neuroinflammation stimulated by LPS by suppressing the TLR4/NF-κB p65 signaling pathway both in vitro and in vivo [15]. Collectively, these data suggest that DYRK1A is a potentially viable target for the treatment of neurodegenerative diseases involving a neuroinflammatory component, such as in PD. This recent finding rationalizes the use of the LPS model for our PK/PD study. In this model, **1c** showed a strong overall anti-inflammatory effect (Figure 2) when **1c** was administered to C57BL/6 mice via the PO route (30 mg/kg, BID) starting 3 days before LPS treatment (i.p., 750 µg/kg BW, for 5 days), significantly (*p* < 0.05) decreasing TNFα accumulation in plasma and in the brain. Finally, we observed that **1c** reduced tau phosphorylation in the hippocampus (Figure 2C).

Because tau is the direct downstream target phosphorylated by DYRK1A, we used the measure of p-tau as a proxy of drug efficacy in tissues. This result suggests that the observed biological activity is due to the inhibition of our target enzyme. When the drug is administered via the IN route, we observed a significant drug level in plasma but a lower level in the brain, which was further confirmed by the lack of efficacy in inhibiting tau phosphorylation. These findings agree with results for other DYRK1A inhibitors [34], as well as with the PK profile of the drug, confirming that 1c administered via the IN route has better peripheral efficacy; however, when administered via the PO route, it achieved better exposure and efficacy in the brain. Dexamethasone, our selected positive control, shows a profile that is similar to that of **1c** administered via the IN route. Like **1c**, dexamethasone achieves high efficacy in plasma but very low efficacy in the brain due to its poor BBB penetration. Overall, **1c** is a potent DYRK1A inhibitor and quite effective in reducing chronic inflammation both systemically and in the brain and could be helpful in the treatment of neuroinflammatory processes such as PD.

#### 2.4.2. MPTP Model

Compound **1c** also demonstrated significant efficacy in the MPTP model for PD (Figure 3). We tested the behavioral effect of 1c in the MPTP model for PD using both IN and PO delivery (both 25 mg/kg, BID). This is a common neurodegenerative disorder model characterized by a progressive loss of dopaminergic (DA) neurons in the striatum [24]. Several studies have shown that oxidative stress, neuroinflammation, and microglial activation play a pivotal role, at least in the progression of PD [35]. The results shown in Figure 3 reveal that the **1c** treatment completely restored the movement behavior of the mice impaired by MPTP in two different behavioral tests, namely the pole test (Figure 3A,B) and the rotarod test (Figure 3C). In these behavioral tests, we did not observe any difference between the two routes of administration. Interestingly, as reported in Figure 3D, tyrosine-hydroxylase-positive cells in the substantia nigra pars compacta region were only protected from MPTP toxicity in the **1c** PO treatment group, while the IN treatment group failed to show any histological difference. As in the LPS model, the PO group showed superior efficacy compared to the IN group, probably because of the better brain/plasma ratio.

## 3. Discussion

DYRK1A is a ubiquitous enzyme that is overexpressed in Down syndrome patients through gene-dosage effects of trisomy chromosome 21 [4]. DYRK1A inhibition offers an attractive approach for the treatment of PD pathologies because of the broad involvement of this kinase in anti-inflammatory pathways (Figure 4). Dementia is recognized as a common component of advanced Parkinson’s disease (PD-D). The combination of LBs, neuroinflammation, AD-type pathologies such as intraneuronal neurofibrillary tangles (NFTs) [36], and extraneuronal neuritic plaques of amyloid β-42 (Aβ_42_) [37,38,39,40,41] is considered to achieve a strong pathological association with PD-D. With the expected increase in global life expectancy and the increasing prevalence of PD, associated dementia is likely to become a prevailing problem. The development of drugs that can target both PD and PD-D is a major unmet need.

DYRK1A is a proline-directed serine/threonine kinase for which many proteins have been shown as substrates [11]. DYRK1A activity may be involved in PD and PD-D pathogenesis because (1) it is robustly expressed in CNS neurons [42]; (2) it increase the clearance of neurotoxic protein aggregates by directly phosphorylating parkin and septin 4 [9,10]; (3) it directly attenuates inflammation by targeting Nrf2, GFAP, and TLR4/NF-κB p65 [15,16,43]; (4) it directly phosphorylates the key protein APP and increases the secretase-mediated cleavage of APP into Aβ peptides [44]; (5) Aβ peptides stimulate DYRK1A expression in a positive feedback loop [45]; (6) DYRK1A is a kinase for which tau serves as a substrate [46]; and (7) its presence is associated with increased phosphorylation of tau [47]. These findings support our hypothesis that the inhibition of DYRK1A activity has a disease-modifying effect and can significantly impact the lives of those with PD and PD-D. We now report the discovery of novel and potent DYRK1A allosteric inhibitors in our ongoing medicinal chemistry effort [48]. Our lead compound in terms of potency, safety, and pharmacological activity is 1c. This compound inhibits DYRK1A, with an IC_50_ of 73 nM, and is characterized by a remarkably clean safety/selectivity profile and improved permeability/bioavailability. The extensive in vitro and in vivo safety studies performed to date have not reveled any concerns. In the PK study, we obtained good bioavailability combined with a good brain/plasma ratio and stable level throughout the 24 h period, making this drug a once-a-day candidate. Owing to its good ADMET properties, **1c** showed excellent activity in several inflammation models. In particular, compound **1c** was found to be efficacious in treating acute inflammation in the LPS model and Parkinson’s symptoms induced by MPTP. These findings indicate that 1c may exert three complementary actions: (i) mitigation of the actions of proteins responsible for neurodegeneration through inhibition of DYRK1A; (ii) direct inflammation attenuation; and (iii) although not directly addressed in the current study, owing to its similarity to EGCG, compound **1c** may be able to increase the clearance of neurotoxic protein aggregates [49]. Therefore, **1c** treatment could simultaneously address several of the predominant underlying pathophysiological aspects of neurodegenerative disorders such as PD and PD-D. All proposed PD drugs based on the inhibition of protein aggregation alone have failed in clinical trials to date. Compound **1c** offers a promising alternative therapeutic strategy. A plan to test this compound in chronic PD models is underway.

## 4. Materials and Methods

### 4.1. Chemistry

General: All solvents and reagents were obtained from commercial suppliers and were used directly without further purification. The reaction progress was monitored on a TLC plate (Merck, silica gel 60 F254, Darmstadt, Germany). ^1^H, ^13^C, and ^19^F NMR spectra were recorded on a Bruker Advance 400 MHz spectrometer using deuterated chloroform or dimethyl sulfoxide (DMSO). Chemical shifts (δ) are reported in parts per million (ppm) up-field from tetramethylsilane (TMS) as an internal standard, and s, d, t, and m are presented as singlet, doublet, triplet, and multiplet, respectively. Coupling constants (J) are reported in hertz (Hz). Liquid chromatography-mass spectra (LC-MS) were recorded on an Agilent (single quad) or Thermo Scientific ion trap. Abbreviations: AA: ascorbic acid; ACN: acetonitrile; DBU: 1,8-diazabicyclo[5.4.0]undec-7-ene; DCM: dichloromethane; DMAP: dimethylamino pyridine; DMF: dimethylformamide; EtOAc: ethyl acetate; MeOH: methanol; PE: petroleum ether; TCEP: tris[2-carboxyethyl]phosphine hydrochloride; TEA: triethyl amine; THF: tetrahydrofuran; TLC: thin-layer chromatography.

#### 4.1.1. Synthesis of (2S,3R)-2-(3,4,5-trihydroxyphenyl)-3,5,7-chromantriol ((−)-gallocatechin, **GC**)

(2R,3R)-2-(3,4,5-trihydroxyphenyl)-3,4-dihydro-2H-chromene-3,5,7-triol (EGC) (50 g) was treated with phosphate buffer (pH 7.2, 0.1 M, 140 mL). The solution was refluxed for 2 h, and after cooling, a white precipitate of gallocatechin was obtained. After filtration, the solid was crystallized with water (500 mL) which afford the desired **GC** as a white solid in good yield with good purity (20 g, 40% yield).

#### 4.1.2. Synthesis of (2S,3R)-5,7-bis(benzyloxy)-2-(3,4,5-tris(benzyloxy)phenyl)chroman-3-ol (**2**)

K_2_CO_3_ (11.30 g, 81.63 mmol, 5.0 eq.) was added to a stirred solution of GC (5.0 g, 16.33 mmol, 1 eq.) in dry DMF (30 mL) and stirred at RT for 0.5 h. Benzyl bromide (9.2 mL, 81.63 mmol, 5.0 eq.) was added dropwise to this mixture at −20 °C. The suspension was slowly warmed to RT and stirred for 24 h. After complete consumption of the starting material, the reaction mixture was filtered through a pad of celite to remove K_2_CO_3_. The celite pad was washed with EtOAc (100 mL). The combined organic phase was washed with cold H_2_O (2 × 50 mL), dried over Na_2_SO_4_, filtered, and concentrated. The obtained residue was purified by flash-column chromatography with EtOAc:Hexane (6:1) to afford the desired intermediate (**2**) (4.5 g, 36% yield) as a white solid. Analytical data: ^1^H NMR (400 MHz, CDCl_3_): δ 7.48 -7.20 (m, 25H), 6.82 (s, 2H), 6.34 (s, 1H), 6.13 (s, 1H), 5.07 (s, 8H), 5.04 (s, 1H), 4.91 (s, 2H), 4.64 (d, J = 7.2 Hz, 1H), 4.03 (bs, 1H), 2.78 (dd, J = 16.0 Hz, 4.8 Hz, 1H), 2.46 (dd, J = 16.4 Hz, 4.8 Hz, 1H).

#### 4.1.3. Synthesis of 3,4,5-tris(benzyloxy)-2-fluorobenzoic acid (**3a**)

H_2_SO_4_ (11.5 mL, 117.564 mmol, 2 eq.) was added to a solution of methyl 3,4,5-trihydroxybenzoate (20 g, 117.564 mmol, 1 eq.) in 200 mL MeOH at 0 °Cm and the reaction mixture was stirred at 80 °C for 22 h. Reaction progress was monitored by TLC. Then, the reaction mixture was concentrated under reduced pressure; the crude residue was diluted with cold water to allow the ester intermediate to precipitate out. The solid was filtered and washed with water, and the wet cake was dried in vacuo to yield the methyl 3,4,5-trihydroxybenzoate as a white solid (20 g, 92% yield). Analytical data: ^1^H NMR (400 MHz, DMSO-d6) δ 9.29 (s, 3H), 6.92 (s, 2H), 3.72 (s, 3H).

K_2_CO_3_ (71.304 g, 515.591 mmol, 5 eq.) was added to a suspension of abovementioned intermediate (19 g, 103.182 mmol, 1eq.) in DMF (200 mL), followed by benzyl bromide (61 mL, 515.591 mmol, 5 eq.) at 0 °C. The mixture was heated to 80 °C for 16 h. After this time, ice was added to the reaction solution to precipitate out the desired product as a solid. The solid was filtered, washed with water, and dried to yield methyl 3,4,5-tris(benzyloxy)benzoate as a white solid (30 g, 64% yield). Analytical data: ^1^H NMR (400 MHz, DMSO-d6) δ 7.43–7.26 (m, 17H), 5.33 (s, 2H), 5.16 (s, 2H), 5.01 (s, 2H), 3.83 (s, 3H).

Selectfluor (46.7 g, 132.013 mmol, 2 eq.) was added to a solution of the abovementioned intermediate (30 g, 66.006 mmol, 1 eq.) in 200 mL of ACN at 0 °C and stirred at RT for 96 h. Reaction progress was monitored by TLC. After this time, the reaction mixture was quenched with a saturated solution of NaHCO_3_, and the product was extracted with EtOAc (3 × 100 mL). The organic layer was washed with brine, dried over anhydrous Na_2_SO_4_, filtered, and concentrated under reduced pressure to yield the crude compound. Purification by flash-column chromatography using 10% EtOAc in hexane afforded the intermediate methyl 3,4,5-tris(benzyloxy)-2-fluorobenzoate as a pale brown solid (7 g, 22% yield). Analytical data: ^1^H NMR (400 MHz, DMSO-d6) δ 7.42–7.28 (m, 16H), 5.33 (s, 2H), 5.16 (s, 2H), 5.14 (s, 2H), 3.81 (s, 3H), 19F NMR (400 MHz, DMSO-d6) δ-134.52.

NaOH (5.9 g, 148.145 mmol, 10 eq.) was added to a solution of the abovementioned intermediate (7 g, 14.814 mmol, 1 eq.) in THF:H_2_O (3:1) (50 mL) and stirred at 80 °C for 6 h. The reaction mixture was concentrated under reduced pressure, the obtained residue was diluted with H_2_O (30 mL), and the product was extracted with EtOAc (2 × 80 mL). The aqueous phase pH was adjusted to <3 with 1N HCl. Then, the mixture was filtered, and the filter cake was dried. The crude compound was purified by flash-column chromatography using 10% MeOH in DCM to obtain the title intermediate (**3a**) as a white solid (3.8 g, 60% yield). Analytical data: ^1^H NMR (400 MHz, DMSO-d6): δ 13.22 (s, 1H), 7.42 (d, J = 1.2 Hz, 2H), 7.44–7.26 (m, 10H), 5.14 (s, 2H), 5.12 (s, 2H), 3.81 (s, 3H).

#### 4.1.4. Synthesis of 3,4,5-tris(benzyloxy)-2,6-difluorobenzoic acid (**3b**)

Selectfluor (77 g, 220.264 mmol, 2 eq.) was added to a solution of methyl 3,4,5-tris(benzyloxy)benzoate (50 g, 110.132 mmol, 1 eq.) in ACN (60 mL) at 0 °C, and the reaction mixture was stirred at RT for 48 h. Reaction progress was monitor by TLC. After this time, the reaction mixture was quenched with cold water, extracted with EtOAc (3 × 100 mL), washed with brine, dried over anhydrous Na_2_SO_4_, filtered, and concentrated under reduced pressure. The obtained crude compound was purified by flash-column chromatography using 5% EtOAc in hexane as an eluent to afford methyl 3,4,5-tris(benzyloxy)-2,6-difluorobenzoate as a yellow solid (0.6 g, 1% yield). Analytical data: ^1^H NMR (400 MHz, DMSO-d6) δ 7.38–7.34 (m, 15H), 5.27 (s, 2H), 5.02 (s, 4H), 3.85 (s, 3H). ^19^F NMR (400 MHz, DMSO-d6) δ-133.38.

LiOH·H_2_O (0.513 g, 12.240 mmol, 10 eq.) was added to a solution of the abovementioned intermediate (0.6 g, 1.224 mmol, 1 eq.) in THF:H_2_O (3:1) (12 mL) and stirred at RT for 16 h. The reaction mixture was concentrated, and the obtained crude material was diluted with H_2_O (30 mL) and extracted with EtOAc (10 mL). The aqueous phase pH was adjusted to <3 with 1N HCl. The obtained solid was filtered and dried to obtain **3b** as a yellow solid (0.352 g, 60% yield). Analytical data: ^1^H NMR (400 MHz, DMSO-d6) δ 13.85 (s, 1H), 7.35–7.33 (m, 15H), 5.15 (s, 2H), 5.02 (s, 4H); ^19^F NMR (400 MHz, DMSO-d6) δ-134.14.

#### 4.1.5. Synthesis of 3, 4-bis(benzyloxy)-2-fluoro-5-methoxybenzoic acid (**3c**)

DBU (1.26 Lit, 8.47 mol, 3.0 eq.) was added to a suspension of methyl 3,4,5-trihydroxybenzoate (520 g, 2.82 mol, 1.0 eq.) in DMF (5.2 L), followed by benzyl bromide (671 mL, 5.65 mol, 2.0 eq.), at 0 °C. The reaction solution was allowed to stir at RT for 48 h. After this time, the reaction mixture was diluted with EtOAc (15.6 L), washed with water (2 × 10.4 L) and brine (10 L), dried over anhydrous Na_2_SO_4_, filtered, and concentrated. The obtained crude compound was purified by column chromatography using DCM in hexane as eluent to yield methyl 3,4-bis(benzyloxy)-5-hydroxybenzoate as a white solid (185 g, 17.9% yield). Analytical data: ^1^H NMR (400 MHz, DMSO-d6) δ 9.77 (s, 1H), 7.46–7.33 (m, 7H), 7.29–7.26 (m, 3H), 7.15 (d, J = 1.6 Hz, 2H), 5.12 (s, 2H), 5.02 (s, 2H), 3.79 (s, 3H).

Selectfluor (170.19 g, 480.76 mmol, 2.5 eq.) was added to a solution of the abovementioned intermediate (70 g, 192.30 mmol, 1.0 eq.) in 700 mL of ethanol at RT and stirred at 80 °C for 30 h. Reaction progress was monitored by TLC. After this time, the reaction mixture was concentrated under reduced pressure, and the obtained crude material was diluted with water (350 mL) and extracted with EtOAc (2 × 700 mL). The combined organic layer was washed with brine (350 mL), dried over anhydrous Na_2_SO_4_, filtered, and concentrated. The obtained crude was purified by column chromatography and eluted with 10% EtOAc in hexane to afford pure methyl 3,4-bis(benzyloxy)-5-hydroxy fluoro derivative as a white solid (15.2 g, 20.69% yield). Analytical data: ^1^H NMR (400 MHz, DMSO-d6) δ 9.94 (s, 1H), 7.48–7.31 (m, 10H), 7.13 (d, J = 6.8 Hz, 1H), 5.14 (s, 2H), 5.00 (s, 2H), 3.80 (s, 3H); ^19^F NMR (400 MHz, DMSO-d6) δ-136.66.

K_2_CO_3_ (36.180 g, 261.78 mmol, 2.5 eq.) was added to a suspension of methyl 3,4-bis(benzyloxy)-5-hydroxy fluoro benzoate (40.0 g, 104.71 mmol, 1.0 eq.) in DMF (400 mL), followed by methyl iodide (13.0 mL, 209.42 mmol, 2.0 eq.), at 0 °C. The mixture was stirred at RT for 3 h. After this time, the reaction mass was diluted with water (800 mL) and EtOAc (400 mL). The organic layer was separated, washed with water (500 mL) and brine (300 mL), dried over anhydrous Na_2_SO_4_, filtered, and concentrated under reduced pressure. The obtained crude compound was triturated with 5% DCM:hexane for 1 h, then filtered. Purification by column chromatography using EtOAc and hexane yielded the desired compound, methyl 3,4-bis(benzyloxy)-2-fluoro-5-methoxybenzoate, as a colorless liquid (18.0 g, 43.4% yield). Analytical data: ^1^H NMR (400 MHz, DMSO-d6) δ 7.47–7.30 (m, 10H), 7.18 (d, J = 6.4 Hz, 1H), 5.11 (s, 2H), 5.03 (s, 2H), 5.01 (s, 2H), 3.84 (s, 6H); ^19^F NMR (400 MHz, DMSO-d6) δ-134.086.

LiOH·H_2_O (12.196 g, 290.40 mmol, 5.0 eq.) was added to a solution of methyl 3,4-bis(benzyloxy)-2-fluoro-5-methoxybenzoate (23.0 g, 58.08 mmol, 1.0 eq.) in THF:MeOH:H_2_O (1:1:1) (240 mL) at 0 °C and stirred at RT for 4 h. After the complete consumption of the starting material on TLC, the solvent was evaporated from the reaction mixture. The obtained residue was diluted with H_2_O (150 mL) and washed with diethyl ether (50 mL). The aqueous layer was acidified with 1N HCl (pH = 3–4), and the formed precipitate was filtered, washed with n-pentane (2 × 50 mL), and dried under vacuum to yield 3,4-bis(benzyloxy)-2-fluoro-5-methoxybenzoic acid **3c** (20.6 g, 92.9% yield) as a white solid. Analytical data: ^1^H NMR (400 MHz, DMSO-d6): δ 13.25 (s, 1H), 7.50–7.30 (m, 10H), 7.19 (d, J = 6.4 Hz, 1H), 5.11 (s, 2H), 5.04 (s, 2H), 3.85 (s, 3H); ^19^F NMR (400 MHz, DMSO-d6) δ-134.263.

#### 4.1.6. Synthesis of benzyl 4,5-bis(benzyloxy)-2-fluoro-3-methoxybenzoate (**3d**)

K_2_CO_3_ (22.4 g, 162.950 mmol, 6 eq.) was added to a solution of 3,4-dihydroxy-5-methoxybenzoic acid (5 g, 27.159 mmol) in DMF (50 mL), followed by benzyl bromide (16 mL, 1135.79 mmol, 5 eq.), at 0 °C. The mixture was heated to 80 °C for 16 h until TLC showed that the reaction was completed. The reaction mixture was diluted with water and extracted with EtOAc. The organic layer was concentrated under vacuum to yield the crude product and purified by flash chromatography using 15% EtOAc in hexane as eluent to yield benzyl 3,4-bis(benzyloxy)-5-methoxybenzoate as a yellow liquid (10.1 g, 82% yield). Analytical data: ^1^H NMR (400 MHz, DMSO-d6) δ 7.43–7.26 (m, 17H), 5.33 (s, 2H), 5.14 (s, 2H), 5.01 (s, 2H), 3.83 (s, 3H).

Selectfluor (17.1 g, 4.400 mmol, 2 eq.) was added to a solution of the abovementioned intermediate (11 g, 24.240 mmol, 1 eq.) in ACN (100 mL) at 0 °C, and the reaction mixture was stirred at RT for 48 h. Reaction progress was monitored by TLC. After this time, the reaction mixture was quenched with cold water and extracted with EtOAc (3 × 100 mL). The organic layer was washed with brine solution, dried over anhydrous Na_2_SO_4_, filtered, and concentrated under reduced pressure. The crude compound was purified by flash-column chromatography using 10% EtOAc in hexane as eluent to yield benzyl 4,5-bis(benzyloxy)-2-fluoro-3-methoxybenzoate as a yellow solid (1.1 g, 9% yield). Analytical data: ^1^H NMR (400 MHz, DMSO-d6): δ 7.42–7.28 (m, 16H), 5.33 (s, 2H), 5.16 (s, 2H), 5.14 (s, 2H), 3.81 (s, 3H); ^19^F NMR (400 MHz, DMSO-d6) δ-134.52.

LiOH·H_2_O (0.88 g, 21.186 mmol, 10.0 eq.) was added to a solution of the abovementioned intermediate (1 g, 2.118 mmol, 1.0 eq.) in THF:H_2_O (3:1) (20 mL). The solution was stirred at RT for 16 h. The reaction mixture was concentrated, and the obtained crude was diluted with H_2_O (30 mL) and extracted with EtOAc (2 × 80 mL). The aqueous phase pH was adjusted to <3 with 1N HCl. The obtained solid was filtered, and the cake was dried. The crude compound was purified by flash-column chromatography using 10% EtOAc in hexane as eluent to obtain **3d** as a white solid (0.502 g, 62% yield). Analytical data: ^1^H NMR (400 MHz, DMSO-d6): δ 13.22 (s, 1H), 7.42 (d, J = 1.2 Hz, 2H), 7.44–7.26 (m, 10H), 5.14 (s, 2H), 5.12 (s, 2H), 3.81 (s, 3H).

#### 4.1.7. Synthesis of 3,4-bis(benzyloxy)-2,6-difluoro-5-methoxybenzoic acid (**3e**)

Selectfluor (77 g, 220.264 mmol, 2 eq.) was added to a solution of methyl 3,4,5-tris(benzyloxy)benzoate (50 g, 110.132 mmol, 1 eq.) in ACN (60 mL) at 0 °C, and the reaction mixture was stirred at RT for 48 h. Reaction progress was monitored by TLC. After this time, the reaction mixture was quenched with cold water, extracted with EtOAc (3 × 100 mL), washed with brine, dried over anhydrous Na_2_SO_4_, filtered, and concentrated under reduced pressure to yield a crude compound. The obtained crude compound was purified by flash-column chromatography and eluted with 5% EtOAc in hexane to obtain methyl 3,4,5-tris(benzyloxy)-2,6-difluorobenzoate as a yellow solid (0.6 g, 1% yield). Analytical data: ^1^H NMR (400 MHz, DMSO-d6) δ 7.38–7.34 (m, 15H), 5.27 (s, 2H), 5.02 (s, 4H), 3.85 (s, 3H); ^19^F NMR (400 MHz, DMSO-d6) δ-133.38.

LiOH.H_2_O (0.31 g, 7.50 mmol, 3.0 eq.) was added to a mixture of the abovementioned intermediate (1 g, 2.50 mmol, 1.0 eq.) in THF:H_2_O (1:1) (20 mL). The solution was stirred at RT for 16 h. The reaction mixture was concentrated to remove THF. Then, the mixture was diluted with H_2_O (30 mL) and extracted with EtAOc (2 × 80 mL). The aqueous phase pH was adjusted to < 3 with 1 N HCl. The obtained solid was filtered, and the filter cake was dried to yield compound **3e** as a white solid (0.85 g, 85% yield). Analytical data: ^1^H NMR (400 MHz, DMSO-d6) δ 13.82 (s, 1H), 7.42 (d, J = 1.2 Hz, 2H), 7.37–7.29 (m, 8H), 5.17 (s, 2H), 5.01 (s, 2H), 3.81 (s, 3H); ^19^F NMR (400 MHz, DMSO-d6) δ-134.65, -135.57. 

#### 4.1.8. Synthesis of 3,4-bis(benzyloxy)-5-(difluoromethoxy)benzoic acid (**3f**)

LiOH·H_2_O (0.25 g, 12.070 mmol, 5.0 eq.) was added to a mixture of methyl 3,4-bis(benzyloxy)-5-(difluoromethoxy)benzoate (1 g, 2.415 mmol, 1.0 eq.) in THF:H_2_O (1:1) (20 mL). The solution was stirred at RT for 16 h. The reaction mixture was concentrated to remove THF. Then, the mixture was diluted with H_2_O (25 mL) and extracted with EtOAc (2 × 30 mL). The aqueous phase pH was adjusted to < 3 with 1N HCl. The obtained solid was filtered, and the filtered cake was dried to yield the compound 3,4-bis(benzyloxy)-5-(difluoromethoxy)benzoic acid as a white solid (0.7 g, 72% yield). Analytical data: ^1^H NMR (400 MHz, DMSO-d6) δ 7.64 (d, J = 1.2 Hz, 1H), 7.46 (d, J = 6 Hz, 3H), 7.39–7.34 (m, 5H), 7.30 (t, J = 2.4 Hz, 3H), 7.11 (s, 1H), 5.15 (s, 2H), 5.01 (s, 2H); 19F NMR (400 MHz, DMSO-d6) δ 80.92, 80.72. 

#### 4.1.9. Synthesis of 3,4-bis(benzyloxy)-5-(difluoromethoxy)-2-fluorobenzoic acid (**3g**)

KOH (0.92 g, 16.48 mmol, 5.0 eq.) was added to a solution of methyl 3,4-bis(benzyloxy)-5-hydroxybenzoate (1.2 g, 3.29 mmol, 1.0 eq.) in ACN:H_2_O (6:4) (10 mL) at room temperature and stirred for 20 min. Then, the mixture was cooled to −78 °C, and diethyl (bromodifluoromethyl)phosphonate was added (2.64 g, 9.89 mmol, 3.0 eq). Then, the mixture was allowed to warm up to RT and stirred for 4 h. Finally, the reaction mixture was diluted with H_2_O (50 mL), neutralized with 1N HCl, and extracted with EtOAc (2 × 100 mL). The combined organic layers were washed with brine (50 mL), dried over Na_2_SO_4_, filtered, and concentrated. The residue was purified by flash-column chromatography using petroleum ether (PE)/EtOAc, 9/1 as eluent to yield methyl 3, 4-bis(benzyloxy)-5-(difluoromethoxy)benzoate (0.48 g, 35% yield) as a yellow solid. Analytical data: ^1^H NMR (400 MHz, DMSO-d6): δ 7.60 (d, J = 2.0 Hz, 1H), 7.52–7.47 (m, 2H), 7.45–7.30 (m, 9H), 7.20 (t, J = 73.6 Hz, 1H), 5.26 (s, 2H), 5.09 (s, 2H), 3.85 (s, 3H). 

Selectfluor (6.15 g, 17.39 mmol, 6.0 eq.) was added to a mixture of the abovementioned intermediate (1.2 g, 2.89 mmol, 1.0 eq.) in ACN (12 mL) at 0 °C and stirred at RT for 1 h. Then, the reaction mixture was warmed to 50 °C and stirred for another 16 h. After completion of the reaction, the reaction mass was cooled to RT, diluted with H_2_O (50 mL), and extracted with EtOAc (2 × 100 mL). The combined organic layers were washed with brine (50 mL), dried over Na_2_SO_4_, filtered, and concentrated. The residue was purified by flash-column chromatography using PE/EtOAc (9/1) as eluent to yield methyl 3,4-bis(benzyloxy)-5-(difluoromethoxy)-2-fluorobenzoate (0.051 g, 4% yield) as a pale yellow solid. Analytical data: ^1^H NMR (400 MHz, CDCl3): δ 7.50 (d, J = 6.4 Hz, 1H), 7.45–7.32 (m, 9H), 6.38 (t, J = 74.0 Hz, 1H), 5.15 (s, 2H), 5.11 (s, 2H), 3.92 (s, 3H). 

LiOH (0.07 g, 2.89 mmol, 5.0 eq.) was added to a solution of the abovementioned intermediate (0.25 g, 0.57 mmol, 1.0 eq.) in MeOH:THF:H_2_O (1:1:1) (6 mL) at 0 °C and stirred at RT for 4 h. After completion of the reaction, the solvent was evaporated under reduced pressure. The obtained solid was diluted with H_2_O (20 mL), acidified with 1N HCl (to pH 2–3), and extracted with EtOAc (3 × 50 mL). The combined organic layers were dried over anhydrous Na_2_SO_4_ and evaporated under reduced pressure to yield **3g** (0.215 g, 89% yield) as a white solid. Analytical data: ^1^H NMR (400 MHz, DMSO-d6): 7.45–7.32 (m, 9H), 7.16 (t, J = 73.2 Hz, 1H), 5.15 (s, 2H), 5.10 (s, 2H). 

#### 4.1.10. Synthesis of 3,4-bis(benzyloxy)-2,6-difluoro-5-isopropoxybenzoic acid (**3h**)

K_2_CO_3_ (5.73 g, 41.20 mmol, 1.2 eq.) was added to a suspension of methyl 3,4-bis(benzyloxy)-5-hydroxybenzoate (10.0 g, 27.470 mmol) in DMF (100 mL), followed by 2-bromopropane (5.08 g, 41.20 mmol, 1.2 eq.), at 0 °C. The reaction mixture was heated to 60 °C for 12 h. After this time, the reaction mass was diluted with water and extracted with EtOAc. The organic layer was evaporated, and the residue was purified by flash chromatography eluted with 25% EtOAc in hexane to yield methyl 3,4-bis(benzyloxy)-5-isopropoxybenzoate as a white solid (8.2 g, 73% yield). Analytical data: ^1^H NMR (400 MHz, DMSO-d6) δ 7.47 (d, J = 1.2 Hz, 1H), 7.45–7.35 (m, 4H), 7.34–7.30 (m, 5H), 7.23 (d, J = 2.0 Hz, 2H), 5.16 (s, 2H), 5.02 (s, 2H), 4.66–4.60 (m, 1H), 3.82 (s, 3H), 1.27 (s, 3H), 1.28 (s, 3H).

Selectfluor (42.7 g, 120.743 mmol, 4 eq.) was added to a solution of the abovementioned intermediate (12.2 g, 30.185 mmol, 1 eq.) in 60 mL ACN at 0 °C, and the reaction mixture was stirred at 60 °C for 32 h. Reaction progress was monitored by TLC. After this time, the reaction mixture was quenched with cold water and extracted with EtOAc (3 × 100 mL). The combined organic layer was washed with brine, dried over anhydrous Na_2_SO_4_, filtered, and concentrated under reduced pressure to obtain a crude compound. The crude compound was purified by flash-column chromatography to yield methyl 3,4-bis(benzyloxy)-2,6-difluoro-5-isopropoxybenzoate as a green solid (1.1 g, 8% yield). Analytical data: ^1^H NMR (400 MHz, DMSO-d6) δ 7.43–7.30 (m, 10H), 5.18 (s, 2H), 5.04 (s, 2H), 4.43–4.28 (m, 1H), 3.56 (s, 3H), 1.24 (s, 3H), 1.16 (s, 3H).

LiOH·H_2_O (0.284 g, 11.300 mmol, 5.0 eq.) was added to a mixture of methyl 3,4-bis(benzyloxy)-2,6-difluoro-5-isopropoxybenzoate (1 g, 2.260 mmol, 1.0 eq.) in THF:H_2_O (1:1) (20 mL). The solution was stirred at RT for 2 h. The reaction mixture was concentrated to remove THF. Then, the mixture was diluted with H_2_O (30 mL) and extracted with EtOAc (20 mL). The aqueous phase pH was adjusted to <3 with 1N HCl. The obtained solid was filtered, and the solid was dried to yield **3h** as a white solid (0.91 g, 94% yield). Analytical data: ^1^H NMR (400 MHz, DMSO-d6) δ 13.82 (s, 1H), 7.43–7.30 (m, 10H), 5.15 (s, 2H), 5.04 (s, 2H), 4.39–4.36 (m, 1H), 1.21 (s, 3H), 1.20 (s, 3H); ^19^F NMR (400 MHz, DMSO-d6) δ 134.17, 134.18, 134.48, 134.48.

#### 4.1.11. General Procedure for Synthesis of Compounds **4a–h**

Oxalyl chloride (5 eq.) and two drops of DMF were added to a stirred solution of **3** (1.2 eq.) in DCM (10 mL) under an inert atmosphere at 0 °C. The reaction mixture was stirred at RT for 3 h. After this time, the reaction mixture was concentrated under reduced pressure to obtain the correspondent acid chloride. The acid chloride was added to a solution of **2** (3.0 g, 3.968 mmol, 1 eq.), DMAP (4 eq.), and TEA (4 eq.) in DCM (10 mL) at 0 °C. Then, the reaction mixture was stirred at RT 16 h. Finally, the reaction was quenched with saturated aqueous NaHCO_3_ solution (5 mL). The organic layer was separated, and the aqueous layer was extracted with DCM (30 mL). The combined organic phase was dried over Na_2_SO_4_, filtered, and concentrated under reduced pressure. The obtained crude compound was purified by flash-column chromatography using EtOAc and hexane as eluent to yield **4a–h**. Please see Appendix A for the detail of compounds **4a–h** synthesis.

#### 4.1.12. General Procedure for Synthesis of Compounds **1a–h**

Pd(OH)_2_ (20 wt. %, 2.0 g) was added to a solution of intermediate **4a–h** (2.0 g) in 20 mL of THF:MeOH (1:1), and the reaction mixture was stirred under a hydrogen atmosphere at RT for 16 h. Then, the mixture was passed through a pad of celite to remove the catalyst. The filtrate was concentrated under reduced pressure. The obtained crude compound was purified by Prep-HPLC to obtain **1a–h** as a white to off-white solid. Please see Appendix A for the detail of compounds **1a–h** synthesis.

### 4.2. Kinase Assays

6xHis-tagged rat-truncated DYRK1A (residues 1-497) was used in the assays, as previously described [28]. DYRK1B was prepared from human DYRK1B isoform p65 as glutathione S-transferase fusion protein, and DYRK2 was prepared from human DYRK2 isoform 1 as 6xHis-tagged protein, as previously described [22]. Kinase preparations were verified by the following immunological and biochemical criteria to ensure their identity before use: (1) immunoreactivity only to the cognate antibody in ELISA (data not shown) and (2) sensitivity to inhibitor AZ-191, which was shown to differentially inhibit different DYRKs [20]. The IC_50_ of AZ-191 for each kinase preparation (Table 1) was determined by the assay protocol described below and found to be consistent with the published IC_50_ values of 88 nM, 17 nM, and 1890 nM for DYRK1A, DYRK1B, and DYRK2, respectively [20].

The IC_50_ of each compound against DYRK1A was determined by an ELISA-based non-radioactive kinase assay as previously described [28] using 6XHis-tagged dynamin 1a proline-rich domain (residues 746-864) as the substrate. The DYRK1A assay protocol can also support DYRK1B and DYRK2 phosphorylation reactions in an enzyme-concentration-dependent manner; therefore, the method was adapted to measuring the IC_50_ of each compound against DYRK1B and DYRK2. For DYRK2, the reactions were performed exactly as described for DYRK1A. For DYRK1B, the assays were similarly conducted but with 30 ng GST-DYRK1B and a kinase reaction time of 60 min.

An ATP competition assay against compound **1c** was performed on DYRK1A as previously described [28]. 

### 4.3. Pharmacokinetic Assays

C57BL/6J mice were given a single dose of drug via the IN or PO route. The formulation for IN delivery was as follows: PEG-400 (12.5% *w*/*w*), HP-β-CD (10% *w*/*w*), and Na_2_EDTA (0.125% *w*/*w*) in water q.s. For PO delivery, we replaced water with saline. The volume for IN delivery was 5 µL/nostril, and the volume used for oral delivery was 10 mL/kg. After sacrifice, blood (500 µL) was collected into K_2_EDTA microtubes and kept on ice all times (must be centrifuged within 30 min of collection). Blood was centrifuged in EDTA vacuum at 4 °C at 15,000× *g* for 4 min; ~250 µL of plasma was collected into polyethylene tubes containing 50 µL of AA/TCEP stabilizing solution (20 mM ascorbic acid and 13 mM TCEP in 50 mM K_2_HPO_4_ buffer; the pH of the solution was adjusted to 6.5 using 2 M NaOH). Then, 200 µL of the collected plasma was mixed with (1) 24 µL of a solution of 10% ascorbic acid and 0.1% EDTA in 40 mM NaH_2_PO_4_; (2) 40 µL of 50 mM sodium phosphate (pH 7.4); (3) 500 units of β-D-glucuronidase type X-A from Escherichia coli (Sigma Chemical Co, St Louis, MO, USA), and (4) 4 units of sulfatase type VIII from abalone entrails (Sigma Chemical Co). The mixture was incubated at 37 °C for 45 min. Plasma was then extracted by the addition of 0.5 mL ACN; the mixture was vortex-mixed for 2 min, then snap-frozen with isopropanol/carbon dioxide dry ice, and the upper layer of ACN was removed. The process was repeated another 2 times. The ACN fractions were pooled into a polyethylene tube kept on ice. The combined fractions were evaporated under a gentle stream of nitrogen at ambient temperature. The residue obtained after evaporation was reconstituted in 200 µL of 75 mM citric acid//25 mM ammonium acetate: ACN (75:25 by vol), and vortexed vigorously for 5 min, and 20 µL of the resulting solution was injected into the LC-MS/MS column.

Drug analysis in tissues: About ~0.4 g of the tissue was homogenized with 1 mL of ice-cold 0.4 M sodium phosphate buffer containing 6 mg of ascorbic acid and 0.5 mg of Na_2_EDTA (final pH of 6.5). After centrifugation at 4 °C at 15,000× *g* for 4 min, the supernatant was collected into polyethylene tubes containing 50 µL of the AA/TCEP stabilizing solution. Then 1 U of sulfatase and 250 U of β-glucuronidase were added to the abovementioned homogenized mixture. Samples were incubated at 37 °C for 45 min, then extracted, dried, and resuspended in a manner similar to that for plasma. Samples were then analyzed by LC-MS/MS.

### 4.4. Pharmacological Studies

#### 4.4.1. LPS-Induced Inflammation Model

This study was performed at BioDuro-Sundia (IACUC Approval Code: BD-202208375, Approval Date: 29 August 2022). C57BL/6J male mice (11–12 weeks old) were housed in a room with automatically controlled temperature (21–25 °C), relative humidity (45–65%), and light–dark (12–12 h) cycles. The mice in each cage were divided into the following treatment groups: (I) i.p. saline group (control); (II) i.p. LPS (750 μg/kg) group; (III) i.p. LPS (750 μg/kg) + dexamethasone (1 mg/kg, PO) group; (IV) i.p. LPS (750 μg/kg) + compound 1c (30 mg/kg, IN, BID) group; and (V) i.p. LPS (750 μg/kg) + compound 1c (30 mg/kg, PO, BID) group. Each group consisted of six male mice. The vehicle for the PO and IN formulations was composed of 12% PEG400, 0.2% Na_2_EDTA, 10% HP-β-CD, and water q.s. Animals were pretreated for 3 days with the drug of choice; then, treatment with LPS commenced on day 0, and on the 5th day of LPS treatment, the drug was administered half an hour before LPS injection. One hour after LPS treatment, the animals were anesthetized with a mixture of ketamine, xylazine 2%, atropine, and saline (4:2.5:1:2.5). The body temperature of mice under anesthesia was maintained by applying a heating blanket and monitored using a rectal thermometer. After anesthesia, the animals underwent cardiac perfusion with ice-cold saline for three minutes (3 mL/minute via peristaltic pump) via the left ventricle. The right atrium was cut as an outflow route. The right hemisphere was post-fixed overnight in 4% PFA in PBS at 4 °C and stored in 1xPBS containing 0.1% (*v*/*w*) sodium azide at 4 °C. The left hemisphere, hippocampus, cortex, midbrain, and brainstem were microdissected, immersed in liquid nitrogen in separate tubes (with one-fifth of the rest of the brain tissue), and stored in microfuge tubes at −80°. TNFα (in both plasma and the hippocampus) was analyzed with ELISA, and the p-tau (AT-8 antibody) level was recorded in the hippocampus and cortex with WB.

#### 4.4.2. MPTP Model

This study was performed at Pharmaron (IACUC approval code: IVP-CNS-06012020; approval date: 10 February 2021). C57BL/6J male mice (8–10 weeks old) were divided into four experimental groups: untreated control group, MPTP-treated group, MPTP + compound **1c** (25 mg/kg, IN) group, and MPTP + compound 1c (25 mg/kg, PO) group. MPTP dissolved in saline was administered via intraperitoneal injections once daily at a dosage of 30 mg/kg/day for 5 consecutive days. The control group was administered intraperitoneal injections of saline.

Treatment with compound **1c** was started one day prior to MPTP treatment and continued for 16 days (twice daily for 15 consecutive days and once on the last day). Compound 1c was administered 1 h before MPTP injection on days 1–5.

The behavioral tests were performed at initiation as the baseline and 3, 6, and 9 days following the last MPTP injection. Pole test: the test consisted of a gauze-taped pole (45 cm high, 1 cm in diameter) with a small cork ball at the top. Mice were placed with their head facing upwards immediately below the ball. Two times were recorded: the time it took for the mouse to turn completely downward (T-turn) and the time it took to descend to the floor (T-total), with a cutoff limit of 60 sec. The test consisted of two trials separated by 10 min intertrial intervals, and the average time was calculated. Mice were pretrained for 3 days before MPTP injection. The rotarod test was used to assess the motor coordination and balance of animals. On the test day, the mice were habituated for 30 min in the test room before testing. The drum was slowly accelerated to a speed of 4–40 rpm for a maximum of 300 s. The latency to fall off the rotarod within this time period was recorded. The test consisted of three trials separated by 10 min intertrial intervals. The mean latency to fall off the rotarod was recorded and used for analysis. Mice were trained for 3 days before MPTP injection. 

Tissue Collection and IHC Analysis: within 2 h following the last dose, the blood of animals was collected via cardiac puncture under isoflurane-induced anesthesia. After blood collection, animals were transcardially perfused with normal saline, followed by buffered formalin fixative. Brains were removed, post-fixed, and embedded in paraffin for subsequent IHC analysis of hydroxylase (TH)-positive cells in the substantia nigra (SN).

## Data Availability

Data is contained within the article and Appendix A.

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
