# Peer review of "Development of Novel Fluorinated Polyphenols as Selective Inhibitors of DYRK1A/B Kinase for Treatment of Neuroinflammatory Diseases including Parkinson’s Disease"

_pharmaceuticals, 2023, doi:10.3390/ph16030443_

Round 1

Reviewer 1 Report

Very minor revisions of the manuscript.

Title: 
Suggested revision of the title into, " Development of novel fluorinated polyphenols as selective inhibitors of DYRK1A/B kinase for treating neuroinflammatory diseases including Parkinson’s disease"
(comma and period removed)

Introduction:
1. line 24: change "above 60" into "above the age of 60"
2. line 86: "Zhou et al." to "Zhou et al. (year)" - place the year of publication

Results and Discussion:
1. Figure 2B: check the spelling of dexamethasone

Materials and Methods:
1. line 561 "Pharmacokinetic assays"
2. For the mouse assays, an ethical approval should be mentioned, which approving body, and the ethics approval code/number if any.
3. line 592: 3.4.1. LPS-induced inflammation model 

Author Response

Thank you for your comments. All the suggestions have been incorporated.

Reviewer 2 Report

This paper is well organized and most of the sections are well written. In this paper authors have discovered a new fluorinated 1c (polyphenol derivative) characterized through improved inhibition against DYRK1A/B enzymes and also improved bioavailability and selectivity. Introduction, methods, results and discussion is adequately summarized, however, there are a few concerns that must be addressed before consideration for publication. 

Abstract is too small and must be elaborated where brief introduction (background), objective of the study, results and discussion should be mentioned. 

Short form should be avoided in the beginning of the sentences throughout the paper.

References should be checked for uniformity.

Author Response

Dear reviewer, thank you for the constructive comment/suggestion, we agree and have addressed all of your suggestions in the revised document

Reviewer 3 Report

It is a manuscript describing new fluorinated polyphenol derivatives. The chemical synthesis is basic and simple and based on previous antecedents of the authors. They have carried out a biological study and an efficient interpretation of them. Therefore, I consider that the manuscript can be published without modifications.

Author Response

Thank you for your review and comments